# A Comparison of Spatial and Phenotypic Immune Profiles of Pancreatic Ductal Adenocarcinoma and Its Precursor Lesions

**DOI:** 10.3390/ijms25052953

**Published:** 2024-03-03

**Authors:** Thomas Enzler, Jiaqi Shi, Jake McGue, Brian D. Griffith, Lei Sun, Vaibhav Sahai, Hari Nathan, Timothy L. Frankel

**Affiliations:** 1Department of Internal Medicine, Division of Hematology/Oncology, University of Michigan, Ann Arbor, MI 48109, USA; 2Department of Pathology, University of Michigan, Ann Arbor, MI 48109, USA; jiaqis@umich.edu; 3Department of Surgery, University of Michigan, Ann Arbor, MI 48109, USA; jmcgue@med.umich.edu (J.M.); briangr@med.umich.edu (B.D.G.); leisun@med.umich.edu (L.S.); drnathan@med.umich.edu (H.N.)

**Keywords:** pancreatic cancer precursor lesions, pancreatic ductal adenocarcinoma, immune microenvironment, immunotherapy

## Abstract

Pancreatic ductal adenocarcinoma (PDAC) is a devastating disease with a 5-year survival rate of 12.5%. PDAC predominantly arises from non-cystic pancreatic intraepithelial neoplasia (PanIN) and cystic intraductal papillary mucinous neoplasm (IPMN). We used multiplex immunofluorescence and computational imaging technology to characterize, map, and compare the immune microenvironments (IMEs) of PDAC and its precursor lesions. We demonstrate that the IME of IPMN was abundantly infiltrated with CD8^+^ T cells and PD-L1-positive antigen-presenting cells (APCs), whereas the IME of PanIN contained fewer CD8^+^ T cells and fewer PD-L1-positive APCs but elevated numbers of immunosuppressive regulatory T cells (Tregs). Thus, immunosuppression in IPMN and PanIN seems to be mediated by different mechanisms. While immunosuppression in IPMN is facilitated by PD-L1 expression on APCs, Tregs seem to play a key role in PanIN. Our findings suggest potential immunotherapeutic interventions for high-risk precursor lesions, namely, targeting PD-1/PD-L1 in IPMN and CTLA-4-positive Tregs in PanIN to restore immunosurveillance and prevent progression to cancer. Tregs accumulate with malignant transformation, as observed in PDAC, and to a lesser extent in IPMN-associated PDAC (IAPA). High numbers of Tregs in the microenvironment of PDAC went along with a markedly decreased interaction between CD8^+^ T cells and cancerous epithelial cells (ECs), highlighting the importance of Tregs as key players in immunosuppression in PDAC. We found evidence that a defect in antigen presentation, further aggravated by PD-L1 expression on APC, may contribute to immunosuppression in IAPA, suggesting a role for PD-L1/PD-1 immune checkpoint inhibitors in the treatment of IAPA.

## 1. Introduction

Pancreatic ductal adenocarcinoma (PDAC) is one of the most lethal cancers [1]. Even if a cancer is resectable, most patients ultimately relapse and die from the disease despite surgery and intense perioperative treatment [2]. A majority of patients present with unresectable or metastatic disease and have a particularly grim prognosis, with a median overall survival of less than 12 months [3]. Chemotherapies have limited efficacy, and accumulating toxicities often prevent the successful continuation of treatment. 

Three main precursor lesions of PDAC have been identified: non-cystic lesion pancreatic intraepithelial neoplasia (PanIN), which is the most common precursor lesion, the cystic lesions intraductal papillary mucinous neoplasm (IPMN), and the less frequently occurring mucinous cystic neoplasm (MCN) (Appendix A) [4,5]. While PanINs cannot be identified using imaging modalities, cystic lesions are radiologically detectable. PanIN is an epithelial neoplasm arising from pancreatic ducts less than 5 mm in diameter, is often found in patients with PDAC (82%), and has recently been detected in normal pancreases in much higher numbers than expected (18/30 samples) [6,7]. IPMN is an epithelial cystic neoplasm that arises in the main pancreatic duct and/or its branches [8]. MCN is a rare neoplasm lined by mucin-producing epithelial cells and has a strong predilection for the female sex [9]. Precursor lesions are divided into three grades based on dysplasia as follows: low, intermediate, and high grade [10]. High-risk precursor lesions, including PanIN or IPMN with high-grade dysplasia, are believed to harbor factors associated with a substantial risk of developing PDAC [11]. When cystic lesions progress to PDAC, the cancer generally remains associated with cysts.

Most cancers evade immune surveillance by recruiting and/or re-programming immune cells that constitute their immune microenvironment (IME) toward an immunosuppressive state [12].

In contrast to many malignancies, immunotherapy with currently available immune checkpoint inhibitors is not effective for PDAC [13]. Accumulating evidence suggests that checkpoint inhibition failure is related to a particularly immunosuppressive IME reinforced by an extensive fibro-inflammatory tissue [14]. PDAC actively mobilize monocytes which differentiate into tumor-associated macrophages (TAMs), myeloid-derived suppressor cells (MDSCs), and regulatory T cells (Tregs) into the IME by secreting granulocyte macrophage-colony-stimulating factor (GM-CSF), chemokine ligand 2 (CCL2), transforming growth factor-β (TGF-β), and other substances [15,16,17]. TAMs are known to directly promote tumor growth and contribute to an immunosuppressive microenvironment by secreting amino acid-degrading enzymes such as arginase, causing arginine depletion that results in the anergy of cytotoxic and helper T cells and the accumulation of Tregs [18,19,20]. MDSCs are not well-defined in human cancers. Tregs are an immunosuppressive subset of CD4^+^ T cells characterized by constitutive expression of the master transcription factor forkhead box P3 (FOXP3) in their nuclei and the immune checkpoint protein cytotoxic T-lymphocyte-associated protein 4 (CTLA-4) on their surface [17,21,22]. Tregs suppress APC function by binding with their inhibitory CTLA-4 to CD80 and CD86 molecules on APCs, preventing the interaction of CD80 and CD86 with the co-stimulatory CD28 receptor on CD4^+^ helper and CD8^+^ cytotoxic T cells [21,22]. Expression of other immune checkpoint proteins such as programmed death-ligand 1 (PD-L1) or programmed death protein 1 (PD-1) on tumor cells, APCs, and T cells further prevents the development of an efficient anti-tumor immune response [14]. 

Better risk stratification of precursor lesions based on biomarkers is urgently needed. This will improve the management of such precursor lesions and help clarify whether close surveillance, resection, or early targeted intervention should be used to prevent progression to cancer [10,11].

In our study, we analyzed the IME of cystic lesions MCN, IPMN, and IPMN-associated PDAC (IAPA), non-cystic lesions PanIN and PDAC, as well as non-neoplastic pancreatic tissues (NNPTs). 

We used multiplex immunophenotyping to co-stain seven markers in formalin-fixed, paraffin-embedded tissues [23,24,25]. This technique allowed us to phenotype the following cell types: ECs, PD-L1-positive ECs, CD8^+^ T cells, CD4^+^ T cells, Tregs, APC, and PD-L1-positive APCs.

## 2. Results

### 2.1. Multispectral Images of Precursor Lesions, Normal Pancreatic Tissue, and Cancerous Lesions

Multiplex immunohistochemical staining and image capture using a multispectral camera allowed us to characterize the immune microenvironments at the cellular level. Increased CD8^+^ T cell infiltration was observed in the microenvironment of MCN (Figure 1A) and even more so in IPMN (Figure 1B). Conversely, a low CD8^+^ T cell infiltration but elevated numbers of Tregs were found in IPMN-associated PDAC (IAPA) (Figure 1C). A moderately increased CD4^+^ T cell infiltration was observed in normal pancreatic tissue (NNPT) (Figure 1D). Elevated CD8^+^ T cell and Tregs infiltrations were found in PanIN (Figure 1E). Overall, low numbers of CD4^+^ and CD8^+^ T cells but highly elevated numbers of Tregs were found in PDAC (Figure 1F). An additional set of multispectral images including a composite image of a PD-L1 stain in PDAC is available in the Appendix A (Appendix A). All images are representative. 

### 2.2. IPMN Are Dominated by CD8^+^ T Cell Infiltration

When we compared cystic lesions among each other and to normal pancreatic tissue, we found that T cell infiltration was highest in IPMN (*p* < 0.0001, comparison: NNPT), while the numbers were lower in IAPA and MCN (Figure 1G). When CD8^+^ T cell to all T cell infiltration was compared, CD8^+^ T cell numbers were highest in IPMN (*p* < 0.0001), significantly lower in IAPA (*p* = 0.0003, comparison: IPMN), and also lower in MCN (Figure 1H). CD4^+^ T cell to all T cell infiltration was relatively high in NNPT and significantly lower in MCN (*p* = 0.0003) and IPMN (*p* < 0.0001) (Figure 1I). Conversely, the infiltration ratio was noticeably higher in IAPA (*p* = 0.0007) when compared with IPMN. 

When we compared non-cystic lesions, we found that T cell infiltration in PanIN was similar to that in NNPT (Figure 1J). T cell infiltration was significantly higher in PDAC than in NNPT (*p* = 0.0250). The ratio of CD8^+^ T cells to all T cells was higher in PanIN than in NNPT (*p* = 0.0266) but lower in PDAC (Figure 1K). CD4^+^ T cell to all T cell infiltration was significantly lower in PanIN (*p* = 0.0027) and PDAC (*p* = 0.00028) than in NNPT (Figure 1L). 

CD8^+^ T cells are involved in immune surveillance and anticancer immune reaction [26,27]. CD8^+^ cytotoxic T cells, also known as cytotoxic T lymphocytes (CTLs), can directly eliminate pathologic cells via the expression and eventual secretion of substances including granzymes, perforin, cathepsin C, and Fas ligand [28,29]. By contrast, CD4^+^ helper T cells play a critical role in initiating and maintaining an immune response [30,31,32]. A high CD4^+^ T cell infiltration but low numbers of CD8^+^ T cells as observed in IAPA could indicate an antigen-processing or -presenting defect in IAPA. Low numbers of CD4^+^ T cells found in cancerous and precursor lesions are consistent with an immunosuppressive environment.

### 2.3. Tregs Infiltration Accumulates with Malignant Transformation

Infiltration of Tregs was low in the microenvironment of cystic lesions MCN and IPMN and slightly elevated in cancerous IAPA (Figure 2A). Correspondingly, the ratio of Tregs to all T cells was low in MCN and IPMN but seemed to be elevated in IAPA (no significance due to the small sample size) (Figure 2B).

Infiltration of Tregs was moderately but significantly up in PanIN (*p* = 0.023) and considerably elevated in PDAC (*p* < 0.0001, comparison: NNPT; *p* = 0.0083, comparison: PanIN) (Figure 2C). Similarly, the ratio of Tregs to all T cells was elevated in PanIN (*p* = 0.003) (Figure 2D). The ratio was much higher in PDAC (*p* < 0.0001, comparison: NNPT; *p* = 0.0001, comparison: PanIN). High levels of Tregs and, more importantly, a high ratio of Tregs to all T cells in cancerous lesions are consistent with a deeply immunosuppressive environment and correlate with a poor prognosis [23,33,34]. 

### 2.4. APCs Are Abundant in IPMN but Not in PanIN or Cancer—High PD-L1 Expression on APCs and Epithelial Cells of PDAC

A significantly elevated APC population was observed in IPMN (*p* = 0.00027) (Figure 3A). In contrast, the population was much lower in IAPA (*p* < 0.0001), supporting the above suggestion of a potential malfunction of the antigen-processing/presenting machinery (APM) in IAPA that may have caused immune escape and led to cancer development [35]. When we analyzed the percentages of PD-L1-positive APCs to all APCs, we found that the ratio was high in IPMN (*p* = 0.0108) and tended to be lower in IAPA (no statistical calculation due to the small sample size) (Figure 3B). 

We found no significant differences in APC infiltrations between NNPT, PanIN, and PDAC (Figure 3C). The PD-L1-positive APC to all APC ratio was low in PanIN (*p* = 0.0397) (Figure 3D). However, the ratio was significantly elevated in PDAC (*p* < 0.0001, comparison: PanIN; *p* = 0.00151, comparison: NNPT). APCs scan and phagocytose pathological antigens [36,37]. They subsequently migrate into draining neighbourhood lymph nodes, where they prime naïve CD8^+^ and CD4^+^ T cells [37,38]. PD-L1 expression on APCs is thought to prevent the establishment of an effective immune response and is associated with adverse cancer outcomes [39,40].

Epithelial cells (ECs) line the inner surfaces of most of the hollow structures in the gastrointestinal tract. ECs can be benign, premalignant, or predominantly malignant and eventually transition from an epithelial to a mesenchymal phenotype [41]. PD-L1-positive EC rates were higher in MCN (*p* = 0.0002) and IPMN (*p* < 0.0001) than in NNPT but significantly lower in IAPA (*p* < 0.0001) than in IPMN (Figure 3E). 

The rates of PD-L1-positive ECs were significantly elevated in PDAC (*p* < 0.0001, comparison: PanIN; *p* = 0.0004, comparison: NNPT), indicating that cancer cells actively hide from immune recognition by expressing PD-L1 (Figure 3F) [42,43]. This is a notable difference from IAPA, where the PD-L1-positive EC to all EC ratio was low. 

### 2.5. High Engagement of CD8^+^ with CD4^+^ T Cells in IPMN but Not in PanIN or IAPA—Malignant Transformation Results in Disruption of the CD8^+^ T Cell Interaction with ECs

The distribution pattern of immune cells correlates with their functional status [25]. A random distribution is considered consistent with non-functional immune cells, as immune cells must be in close proximity to each other to exert their function [25,44,45]. To determine cellular engagement within the microenvironment, we selected a circular area with a radius of 15 μm around each T cell and a radius of 40 μm around each APC or EC (Appendix A) [25]. The median frequency of cells within this engagement zone was calculated as the percentage of engaged cells relative to all cells.

When we analyzed the engagement of CD4^+^ and CD8^+^ T cells in cystic lesions, we found that the percentage of CD4^+^ and CD8^+^ T cells in close proximity was high in IPMN (*p* < 0.0001, comparison: NNPT) but low in IAPA (*p* = 0.0383, comparison: IPMN) (Figure 4A). Neighbourhood analysis of CD4^+^ T cells and APCs revealed no significant differences in engagement in MCN and IPMN compared to NNPT, but there was a significant drop in engagement in IAPA (*p* = 0.001, comparison: NNPA; *p* = 0.0005, comparison: IPMN) (Figure 4B). This further strengthens our hypothesis of a defect in the APM in IAPA. Next, we performed neighbourhood analysis of CD4^+^ T cells and PD-L1-positive APCs. IPMN seemed to score higher than NNPT, while MCN was lower and IAPA was the lowest (no statistical calculation due to the small sample size) (Figure 4C). Neighbourhood analysis of CD8^+^ T cells and Tregs revealed similar percentages of engagement for MCN, IPMN, and IAPA (Figure 4D). Analyses of CD8^+^ T cells and APCs demonstrated a significant drop in interactions in IAPA compared with IPMN (*p* = 0.0013) (Figure 4E). In addition, the interaction between CD8^+^ T cells and PD-L1-positive APCs seemed to be much lower in IAPA than in IPMN (no statistical calculation due to the small sample size) (Appendix A). Not surprisingly, the interaction between CD8^+^ T cells and ECs was low in IAPA (*p* < 0.0001, comparison: NNPT and IPMN) (Figure 4F). The interaction was also relatively low in MCN (*p* = 0.0042). The interaction between CD8^+^ cells and PD-L1-positive ECs was particularly low in IAPA (*p* < 0.0001, comparison: IPMN) (Appendix A). Our findings demonstrate that despite low percentages of PD-L1 expression on APCs and ECs in IAPA, PD-L1 expression on those cells resulted in a marked disruption of the engagement with CD4^+^ and CD8^+^ T cells.

When we analyzed non-cystic lesions, we found that the percentages of close proximities of CD4^+^ and CD8^+^ T cells were relatively low in PanIN but elevated in PDAC (*p* = 0.0002, comparison: NNPT; *p* = 0.0469, comparison: PanIN) (Figure 4G). Neighbourhood analysis of CD4^+^ T cells and APCs revealed a lower engagement in PanIN (*p* = 0.012, comparison: NNPT) but higher engagement in PDAC (*p* = 0.0227, comparison: PanIN) (Figure 4H). Neighbourhood analysis of CD4^+^ T cells with PD-L1-positive APCs revealed a considerably higher engagement in PDAC compared with PanIN (*p* < 0.0001) (Figure 4I). Neighbourhood analysis of CD8^+^ T cells and Tregs revealed a significantly higher interaction in PDAC (*p* < 0.0001, comparison: NNPT), which was lower in PanIN (*p* = 0.029, comparison: PDAC) (Figure 4J). The close proximity of CD8^+^ T cells to Tregs leads to the exhaustion and anergy of CD8^+^ T cells [46,47]. The percentages of CD8^+^ T cell interactions with APCs were lower in PanIN than in NNPT and PDAC (*p* = 0.0451, comparison: PanIN) (Figure 4K). The interaction between CD8^+^ cells and PD-L1-positive APCs was low in PanIN (*p* = 0.0043, comparison: NNPT) and significantly higher in PDAC (*p* < 0.0001, comparison: PanIN) (Appendix A). The engagement of CD8^+^ T cells and ECs was very low in PDAC (*p* = 0.0002, comparison: NNPT; *p* = 0.0349, comparison: PanIN) (Figure 4L). The engagement of CD8^+^ T cells with PD-L1-positive ECs was low in PanIN but higher in PDAC (*p* = 0.0022, comparison: PanIN) (Appendix A). In conclusion, the IMEs of IPMN and PanIN differ fundamentally. While T cell interactions among each other and with APCs are high in IPMN, these interactions are much lower in PanIN. Cancerous transformation is accompanied by a dramatic decrease in the engagement of CD8^+^ T cells with ECs and, to a lesser degree, with APCs. Conversely, there was a substantial interaction between CD8^+^ cells and PD-L1-positive APCs and ECs in PDAC but not in IAPA, marking another difference between PDAC and IAPA. 

### 2.6. Engagement of APCs with ECs Is Low in Precursor Lesions and Cancer

The interaction between APCs and ECs was low in cystic lesions MCN (*p* = 0.0006) and IPMN (*p* = 0.0169) compared with normal tissues. The cancerous transformation of IPMN to IAPA was accompanied by a further drop in interaction (*p* = 0.0224) compared with IPMN. P was 0.0002 when compared with NNPT (Figure 5A). The interaction between APCs and PD-L1-positive ECs was relatively low in MCN and IPMN, whereas it was exceptionally low in IAPA (*p* < 0.0001, comparison: IPMN) (Figure 5B), bolstering our suggestion of PD-L1 expression having a substantial impact on immunosuppression in IAPA. 

The interaction between APCs and ECs was lower in PDAC than that in NNPT (*p* = 0.0363) (Figure 5C). The interaction between APCs and PD-L1-positive ECs was not significantly different in NNPT and PDAC, while it was significantly lower in PanIN (*p* = 0.0051) than in PDAC (Figure 5D).

### 2.7. Negative Impact of Tregs on the Engagement of CD8^+^ T Cells with ECs in PDAC but Not in Precursor Lesions—High Degree of Cellular Mixing in the IME of PDAC Indicative of a Non-Functional Immune Environment

We investigated the impact of the numbers of Tregs in the IME on the engagement of CD8^+^ T cells with ECs. In PDAC, we found that a high ratio of Tregs to all T cells was accompanied by an impressive decrease in the interaction between ECs and CD8^+^ T cells (Figure 6A). No such decrease in interaction was observed in PanIN (Figure 6C) or the other precursors. High numbers of APCs in proximity to CD4^+^ T helper cells correlated positively with the proximity of ECs and CD8^+^ T cells (Figure 6B). This indicates that there is a direct relationship between the activity of APCs and the proximity of CD8^+^ cytotoxic T cells to cancer cells. 

We utilized the G-function to characterize the population-level mixing of different cell types in the IME [25,48]. The rate of increase in the G-function served as a surrogate to measure the degree of cellular mixing, and the area under the curve (AUC) was used to compare differences in cellular mixing at a fixed radius from individual cells. A high AUC correlates with a high degree of mixing, which is consistent with a chaotically organized, non-functional immune environment. We calculated the AUCs for CD8^+^ T cells and Tregs. We observed a significantly higher population mixing in PDAC (*p* < 0.0001) than in NNPT. Mixing was lower in IAPA and all precursor lesions (Figure 6D).

## 3. Discussion

In our work, we sought to map and compare the immune microenvironments of cystic and non-cystic pancreatic cancer precursor lesions, as well as of PDAC, IAPA, and normal pancreatic tissue. 

We found a high CD8^+^ T cell infiltration in IPMN, likely reflecting an active and dynamic immune environment [37,49]. Such an immune environment includes the presentation of “pathogenic” antigens to CD8^+^ T cells, which become stimulated and differentiate into CTL within the lymphoid tissue before migrating to the source of the pathogenic antigen [37]. Unsurprisingly, we found that cancerous transformation was accompanied by a dramatic decrease in CD8^+^ T cell infiltration. 

On the other hand, Tregs infiltrations were significantly increased in the IME of cancerous lesions. A moderately increased ratio of Tregs to all T cells was also found in PanIN, but not in other precursor lesions, pointing toward a Tregs-mediated immunosuppressive microenvironment in PanIN [50]. Immunosuppression in PanIN is well known, and on the cell signaling level, activating KRAS mutations seems to promote this immunosuppressive state [51,52]. KRAS mutations are present in up to 90% of early PanIN; they are also present in other precursor lesions, albeit less frequently [5,6,53]. Based on our findings, we hypothesize that targeting Tregs, for example, with an anti-CTLA-4 checkpoint inhibitor in biopsy-proven high-risk PanIN could be a useful therapeutic intervention to restore immune surveillance and eventually reverse the condition or prevent progression to invasive cancer (Figure 1) [21,54,55]. There is some controversy about depleting Tregs causing increased myeloid cell recruitment based on mouse models of PanIN and PDAC [56]. Thus, simultaneous blockage of the common receptor CCR1 of pro-inflammatory chemokines CCL3, CCL6, and CCL86/8 could be reasonable to prevent excessive influx of pro-inflammatory and immunosuppressive myeloid cells into the IME [56,57]. 

APC infiltrations were elevated in cystic precursors, particularly in IPMN, but not in non-cystic lesions. This is concordant with our observation of elevated CD8^+^ T cell infiltrations in cystic precursors. In IPMN, however, a relatively high percentage of APCs exhibited PD-L1 expression. High PD-L1 expression on APCs can prevent a successful immune response by inducing exhaustion and apoptosis of CD8^+^ T cells [58,59]. Thus, it seems that in IPMN immune surveillance mediated by high CD8^+^ T cell and APC numbers is tapered by elevated PD-L1 expression on APCs. This raises the important question of whether treating high-risk IPMN with anti-PD-1/PD-L1 checkpoint inhibitors could restore immune surveillance and prevent their progression to PDAC (Figure 1) [60]. 

Neighbourhood analysis of CD4^+^ T cells, CD8^+^ T cells, and APCs revealed high cellular engagements in IPMN, while engagements in IAPA were very low. These findings suggest that IPMN may possess a highly active APM, which does not seem to be the case in IAPA. Rather, the low engagement of CD4^+^ T cells with CD8^+^ T cells and APCs in IAPA points toward an acquired defect in the APM. Loss or malfunction of antigen presentation is an important mechanism of immune escape in cancers [61]. In this regard, the human leukocyte antigen (HLA) class I proteins play a crucial role in presenting tumor antigens to T cells [61,62]. 

Similar to the high numbers of CD8^+^ T cell interactions with APCs in IPMN, we found relatively high percentages of interactions between ECs and CD8^+^ T cells in those precursors, which decreased dramatically in IAPA. A low percentage of interactions between CD8^+^ T cells and ECs was also observed in PDAC. The interaction between CD8^+^ T cells and PD-L1-positive ECs was exceptionally low in IAPA but not in PDAC. Also, neighbourhood analysis of APCs with PD-L1-positive ECs revealed a very low interaction profile in IAPA but not in PDAC: only 4.8% (mean, ±SD 10.0%) of APCs interacted with PD-L1-positive ECs in IAPA while engagement was 31.1% (mean, ±SD 3.4%) in PDAC. Our findings point toward a substantial negative impact of the PD-1/PD-L1 axis on cellular interactions above all in IAPA but less in PDAC. Based on these observations, we hypothesize that despite low rates of PD-L1 expression on APCs and ECs in IAPA, the addition of PD-1/PD-L1 immune checkpoint blockade to standard-of-care chemotherapy protocols may improve therapeutic efficacy in IAPA. To date, immunotherapy for pancreatic adenocarcinoma has been largely unsuccessful; however, studies have not focused on immunotherapy for cystic pancreatic cancers. A major goal of immune research is to identify subgroups of PDACs that respond to immunotherapy-based regimens [63]. Our findings indicate that IAPA may represent such a subgroup.

Neighbourhood analysis of CD8^+^ T cells and Tregs revealed a significantly increased percentage of engagement in PDAC compared with precursor lesions. This was much less the case in IAPA and seems to mark another important difference between IAPA and PDAC. While Tregs seem to be key players in maintaining immunosuppression in PDAC, a defect in the APM seems to be an important mechanism of immunosuppression in IAPA. 

In PDAC, a high ratio of Tregs to all T cell infiltrations was accompanied by an impressive decrease in the interaction between ECs and CD8^+^ cytotoxic T cells, suggesting the absence of a CTL-mediated anti-tumor response. We did not observe such a correlation in PanIN or other lesions. Our findings highlight the importance of Tregs in maintaining an immunosuppressive environment in PDAC. 

We also demonstrated that in PDAC an active APM mirrored by the close proximity of CD4^+^ helper T cells to APCs correlated positively with the proximity of CTL to cancer cells. In summary, our observations emphasize the dual role of APCs in pancreatic cancer as follows: (1) to generate an efficient anti-tumor immune response and (2) to block/prevent an immune response by the expression of immune checkpoint proteins and/or a defect in the APM.

Cellular mixing, as measured by G-function analysis, is another approach to identifying functional cellular interactions [25,64]. We observed a high degree of mixing between CD8^+^ T cells and Tregs in PDAC. Mixing was clearly less in IAPA, PanIN, and IPMN and least in NNPT. Our findings are consistent with a completely disorganized and non-functional immune microenvironment in PDAC.

Besides the small sample sizes of certain tissues, another limitation of our study is that we focused on cellular interactions between T cells and their subsets, i.e., APCs and ECs, but did not include other cells due to technical limitations. Studying the influence of other cells that constitute the IME such as TAM, MDSC, and neutrophils will further contribute to the understanding of cellular interactions in the IME. 

## 4. Materials and Methods

Patients. We analyzed cohorts of patients with surgically resected MCN, IPMN, IAPA, PanIN, PDAC, and NNPT. After a review of the whole slides by a trained gastrointestinal pathologist, three 0.6 mm diameter cores were collected from the tissue blocks for inclusion in a tissue microarray (TMA). Appendix A gives a detailed overview of the sample sizes represented in the different experiments. 

Multiplexed IHC staining. Slices (5 μm) were cut from the TMA onto charged slides, and the slides were treated as previously described [25]. An Opal 7 manual kit (Akoya Biosciences) was used according to the manufacturer’s instructions. First, the slides were stained with antigen-specific primary antibodies. We used the following antibodies: CD3 (Agilent Dako AO452 (polyclonal)), CD8 (M5390 (SP239), Spring Biosciences, Pleasanton, CA, U.S.A.), FOXP3 (12653 (D608R), Cell Signaling Technology, Danvers, MA, U.S.A.), CD163 (NCL-L-CD163 (10D6), Leica Biosystems, Deer Park, IL, U.S.A.), PD-L1 (13684 (E1L3N), Cell Signaling Technology, Danvers, MA, U.S.A.), and pancytokeratin (M3515 (AE1/AE3), Agilent Dako, Santa Clara, CA, U.S.A.). Phenotypes were assigned as follows: T cells (CD3^+^), Tregs (CD3^+^CD8-FOXP3^+^), helper T cells (CD3^+^CD8-FOXP3-), cytotoxic T cells (CD3^+^CD8^+^), ECs (PanCK+), and APCs (CD163^+^). After a primary antibody was applied, Opal Polymer (secondary antibody) was added in a second step. The application of Opal Tyramide Signal Amplification (TSA) then created a covalent bond between the fluorophore and the tissue at the HRP site as described in [25]. 

Multispectral imaging. Imaging was completed using the Vectra Quantitative Pathology Imaging System. One image per core was captured at ×20 magnification. All cube filters were used for imaging (DAPI, CY3, CY5, CY7, Texas Red, Qdot, Invitrogen, Waltham, MA, USA). The incorporated saturation protection was set to an exposure time of 250 ms.

Image analysis. Images were analyzed using inForm cell analysis software (v 2.4.1; Perkin Elmer, Waltham, MA, USA) as described in [25]. The fluorescent intensity score was determined for PD-L1, CD4, CD8, and FoxP3 using R programs, together with the original cell phenotypes produced by inForm (T cells, APCs, ECs). Final multiplex fluorescent composite images were reviewed by an experienced gastrointestinal pathologist to confirm the accuracy of staining and phenotyping. 

Spatial analysis. Nearest neighbour and cell counts within radius r were calculated using Python 3.7 based on data obtained with inForm. Cell-to-cell distances were calculated. Phenotypes of interest were selected, and a distance matrix from one cell phenotype population to another was generated. Nearest neighbour distances and cell counts within a specific radius were calculated. Individual procedural steps and representative raw images are shown in Appendix A. A G-function was calculated to quantify the spatial relationships and interactions between two or more cell types as described in [25,48,65]. The G-function is a mathematical formula that computes the probability of a reference cell phenotype having a non-reference phenotype within a certain distance. 

Statistics. All statistical analyses were performed using JMP Pro 14 software. Differences in phenotype, distance, and engagement were evaluated by a 2-tailed Student’s *t*-test or ANOVA. For non-normally distributed data, the nonparametric Wilcoxon rank-sum test was used. We used the Kruskal–Wallis rank sum test for a one-way analysis of variance. Categorical variables were analyzed using Fisher’s exact test, and statistical significance was set at *p* ≤ 0.05.

Ethical statement. All procedures performed involving human participants were in accordance with the ethical standards of the University of Michigan research committee and with the 1964 Helsinki Declaration and approved by the University of Michigan Institutional Review Board (HUM00098128) on 3/10/2015. 

Informed consent statement. Informed consent was obtained from all subjects involved in this study.

## 5. Conclusions

We analyzed the immune microenvironments of cystic precursor lesions MCN-, IPMN-, and IPMN-associated PDAC (IAPA), as well as non-cystic precursor lesions PanIN and PDAC, using multiplex immunophenotyping. We identified different types of immunosuppression in PanIN and IPMN: While immunosuppression in PanIN seems to be mediated by elevated infiltrations of Tregs, immunosuppression in IPMN seems to be mediated by high expression of PD-L1 on APCs and ECs. Based on our findings, we hypothesize that prophylactic treatment of high-risk PanIN and IPMN could prevent progression to cancer. We propose treatment as follows: (1) PanIN, anti-CTLA-4, eventually combined with an antagonist targeting the pro-inflammatory cytokine receptor CCR1, and (2) IPMN, anti-PD-L1/PD-1. We further identified IPMN-associated PDAC (IAPA) as a potential candidate for treatment with anti-PD-L1/PD-1 combined with conventional treatment modalities, as immunosuppression in IAPA seems to be the result of a defect in the APM, further aggravated by PD-L1 expression on APCs and ECs, and not primarily mediated by Tregs as in PDAC.

## Data Availability

Data published in this report are available upon request from the corresponding author.

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
