# Peer review of "A Comparison of Spatial and Phenotypic Immune Profiles of Pancreatic Ductal Adenocarcinoma and Its Precursor Lesions"

_ijms, 2024, doi:10.3390/ijms25052953_

Round 1
Reviewer 1 Report
Comments and Suggestions for Authors
This study performed multiplex immunohistochemistry in cystic lesion-derived MCN, IPMN and IAPA, non-cystic lesion-derived PanIN and PDAC, and normal pancreas tissues to detect the distribution of CD4+ and CD8+ T cells, Tregs, APCs and ECs, as well as PD-L1+ APCs and ECs. Together with the neighboring analysis and G-function test, this study revealed distinct characteristics of immune microenvironment in each tumor type and the authors proposed that possible preventive therapeutic approaches, immune checkpoint inhibition to target Tregs in combination with CCR1 for PanIns to prevent PDAC, and PD-1/PD-L1 inhibition for IPMN to prevent IAPA. It is a quite interesting and informative point of view. Several points should be added and are to be addressed.
1) Firstly, the authors should clearly document how many tissue samples were examined in each tissue type.
2) Secondly, the authors should clearly document which antibodies (company name and clone name) were used. Moreover, the authors should clearly document how they determined the epithelial cells, APCs and Tregs. Which antigens were detected by which antibodies?
3) Representative image of PD-L1 staining should be shown. In addition, representative raw image.of assessing the neighborhood analysis should also be shown.
4) Figure S1 should be mentioned in the text.
5) In the Results 2.1, they mentioned about Tregs. Figure 2 data should be included in Figure 1.
6) In line 157, the description should be “tended to be lower in IAPA” because of no statistical difference.
7) Abbreviated words should be spelled out at the first-time appearance.
8) For better understanding, Scheme 1 had better include PDAC and IAPA.
9) In lines 334-344, the authors described repeatedly low positive rates in the neighborhood analysis in IAPA, so, it is a bit difficult to understand the next sentence that PD-1/PD-L1 inhibition is to be applied to IPMN.
10) The authors described a bit about TAMs and MDSCs, but this study focused on T cells and APCs, did not examine TAMs, MDSCs, and neutrophils. This might be limitation of this study and should be discussed about it.
Comments on the Quality of English Language
Abbreviated words should be spelled out at the first-time appearance.
There are some typos.
Reviewer 2 Report
Comments and Suggestions for Authors
In this manuscript, the authors present a study where immune markers are analyzed across different phenotype of pancreatic cancer using multi-immunohistochemistry. Although the study is interesting, there is a lack in the description of the cohort and methods used. My main concern is about the size of the patients’ cohort and surface of tissue analyzed. No information are provided. The authors did not provide any information how the spatial analysis is performed. Moreover, I suggest to use additional mIHC images to support their findings and select only a few boxplots moving the rest in supplementary material. Multivariate analysis could be used to better visualize the different immune profile of the PDAC phenotypes.
Few typos are noted:
Rows 18, 22, 198: please change IMPN with IPMN
Row 46: change pancreata with pancreas
Round 2
Reviewer 2 Report
Comments and Suggestions for Authors
The authors replied exhaustively to my comments.